# Slurry Leakage Channel Detection and Slurry Transport Process Simulation for Overburden Bed Separation Grouting Project: A Case Study from the Wuyang Coal Mine, Northern China

Tianhao Kou [1], Shuaixin Wen [1], Wenping Mu [1,*], Nengxiong Xu [1], Zexin Gao [1], Zhongxiang Lin [2], Yankui Hao [2], Weitao Yan [3] and Honglei Liu [4]

[1] School of Engineering and Technology, China University of Geosciences (Beijing), Beijing 100083, China
[2] China Coal Geological Group Co., Ltd., Beijing 100040, China
[3] School of Surveying and Land Information Engineering, Henan Polytechnic University, Jiaozuo 454003, China
[4] National Coal Mine Water Hazard Prevention Engineering Technology Research Center,
China University of Mining and Technology (Beijing), Beijing 100083, China
* Correspondence: muwp@cugb.edu.cn; Tel.: +86-186-1081-4064

**Abstract:** The 8006 working face at the Wuyang Coal Mine adopts grout injection into bed separation technology for surface subsidence control. Surface grout leakage occurred during the grout injection into the bed separation process of this working face. Grout leakage has adverse effects on the grouting filling effect, grouting cost and the environment. To determine the grout leakage channels and the slurry transport process, and to provide a theoretical basis for slurry leakage prevention and control, this paper first used 3D seismic exploration technology to identify the fault distribution characteristics of the study area, and then used COMSOL Multiphysics to carry out the numerical simulation of the grout transport process. The conclusions are as follows. Fifteen normal faults were identified in the vicinity of the 8006 working face. Among all the faults, the F1, F11, F18, F19 and F27 faults penetrate the surface and are the main channels for the grout to run to the surface. Based on the distribution characteristics of the faults and the spatial location relationship among the bed separation, faults and grout leakage points, the theoretical analysis of the leakage causes of each grout leakage point was carried out, and the main leakage channels of the grout injection into bed separation were proposed to be the bed separation and faults. The results of the numerical simulation of grout transport show that, as the permeability of the bed separation space and fault is much better than that of the surrounding rock, during the grout injection process the grout diffuses through the bed separation and fault in turn, and finally to the surface, where leakage occurs. The simulation results confirm that the main leakage channels for the grout are bed separation and faults.

**Keywords:** grout injection into bed separation; 3D seismic exploration; grout leakage channels; grout leakage simulation

## 1. Introduction

Surface subsidence during coal mining can cause serious impacts and damage to surface buildings and structures, so controlling surface subsidence has become an important research topic. Grout injection into bed separation is a new coal mining subsidence reduction technology developed for mining coal under buildings that can control the collapse and deformation of the overburden rock layer to the maximum extent, thus reducing the deformation of the ground surface and effectively protecting the surface buildings [1–4]. The technology is based on the key strata theory [5]. The slurry is injected through ground boreholes into the bed separation space beneath the key strata to prevent the key strata from breaking up, thereby controlling the surface settlement and protecting the safety of the surface buildings. In recent years, grout injection into bed separation technology has been greatly developed in terms of grouting theory research and engineering applications [6–12].

In addition, the flow and transport of slurry is of great importance for the borehole layout, the determination of grout and grout solids distribution, the control of grouting volume, and the prevention of surface grout leakage. It is also the key research content of overburden separation grouting technology.

In the field of fracture grouting in rock and soil masses, many scholars have carried out detailed studies on grout transport and flow in fissures [13–15], and their research methods and results have provided good references for the study of the flow and transport of grout in bed separation. At present, scholars from various countries have mainly studied the grout flow of rock fractures from three aspects: theoretical models, physical model tests, and numerical simulations. In terms of theoretical models, due to the complexity of rock fracture distribution, the analytical solutions mainly focus on the channel flow and radial flow in single fractures. Based on the assumption of lubrication approximation, some scholars have derived analytical solutions for the flow of fluids with different rheological properties in single fissures [16–20]. Model tests and numerical simulations have been widely used to verify theoretical models [16–18] and to explore the complex flow of grout fluids [21,22]. To a certain extent, the above research results have contributed to the research process of grout transport and flow. However, cement-based grout is generally used for fracture grouting in rock and soil masses, and most grouted fractures are small-scale primary fractures in rock masses. In contrast, grout injection into bed separation technology in most cases uses fly ash grout, and the scale of the filled bed separation space is much larger than that of the primary fractures in the rock mass. Therefore, the flow law of fracture grouting in rock and soil masses is not fully applicable to grout injection into bed separation.

Due to the strong concealment of grout in bed separation space, scholars from various countries have conducted limited research on its transport and flow law. At present, some scholars have studied the flow and distribution of grout in bed separation space, calculated the injection-production ratio and grouting amount, and established theoretical models [23–26]. These studies have served as a good guide for grouting engineering applications. Dayang Xuan [27] used a self-developed visual experimental system of bed separation grout injection to study the flow, pressure distribution, consolidation, and fill thickness of fly ash slurry in overburden bed separation. In addition, numerous studies have confirmed that faults are closely related to mine water inrush [28–32]. The fault destroys the integrity of the coal seam floor and significantly shortens the vertical distance between the bottom aquifer and the coal seam. Groundwater is prone to complex physical, chemical and mechanical effects in fault zones. All of these characteristics lead to faults being the main channel for water inrush from the coal mine floor [33]. Analogously, the fault is also the main channel for the dispersion of the slurry and the precipitated water.

In summary, there are few studies specifically on the slurry flow of an overburden bed separation grouting project. The research on the slurry flow and slurry transport channel in the overburden bed separation grouting project is not clear. On the other hand, most of the existing research in the field of both fracture and bed separation grouting is focused on the flow of slurry in a single fracture or a single bed separation space, and very little research has been carried out on the surface grout leakage caused by the longitudinal transport of slurry along a channel such as a fault. However, in bed separation grout injection projects, surface grout leakage has serious negative impacts on the bed separation grout injection filling effect, the grouting cost, and the environment. Therefore, there is an urgent need to fully investigate the grout leakage channels and slurry transport processes in overburden slurry injection.

Based on the research background of the overburden bed separation grout injection project of Wuyang Coal Mine in China, this paper analyses the slurry escape channel and conducts a numerical simulation of the slurry transport process from the injected bed separation space to the surface. The following results are obtained in this paper. Fifteen normal faults were identified in the vicinity of the 8006 working face. Among all the faults, the F1, F11, F18, F19 and F27 faults penetrate the surface and are the main channels for the grout to run to the surface. Based on the distribution characteristics of the faults and the

spatial location relationship among the bed separation, faults and grout leakage points, the theoretical analysis of the leakage causes of each grout leakage point was carried out, and the main leakage channels of the grout injection into bed separation were proposed to be the bed separation and faults. The results of the numerical simulation of grout transport show that, as the permeability of the bed separation space and fault is much better than that of the surrounding rock, during the grout injection process, the grout diffuses through the bed separation and fault in turn and finally to the surface, where leakage occurs. The simulation results confirm that the main leakage channels for the grout are bed separation and faults. The research results of this paper are of great significance in guiding engineering problems such as the borehole layout, the determination of grout and grout solids distribution, the control of grouting volume, and the prevention of surface grout leakage.

## 2. Study Area

The Wuyang Mine is located in the western part of Changzhi city, Shanxi Province, at the western foot of the Taihang Mountains, with the Shangdang Basin to the south. The topography of the study area is dominated by the low hilly terrain of the Loess Plateau, with the overall terrain showing a high southwest and low northeast elevation. Gullies have developed in the study area. Most of the area is covered by loess, with sporadic outcrops of Middle Ordovician strata and Permian strata. The study area has a temperate monsoon climate with an average annual temperature of approximately 9 °C, an average annual precipitation of approximately 600 mm, and an average annual evaporation of approximately 1400 mm. The location of the study area and the range of the seismic survey area are shown in Figure 1.

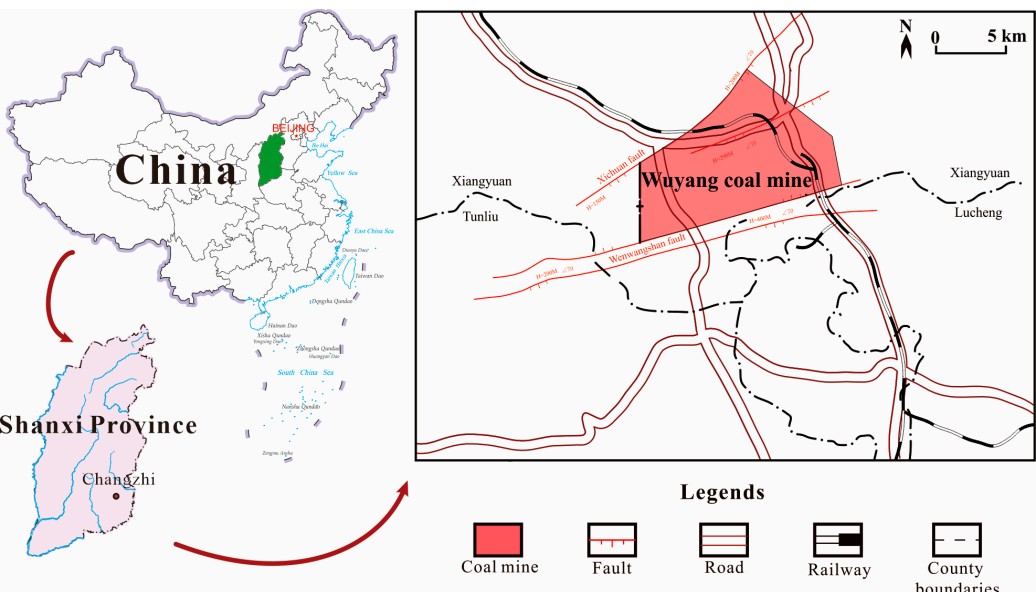

**Figure 1.** Location of the study area.

The strata in the study area from old to new are the Upper Majiagou Formation and Fengfeng Formation of the Ordovician Middle Series, Benxi Formation of the Carboniferous Middle Series, the Upper Taiyuan Formation of the Carboniferous Upper Series, the Shanxi Formation and Lower Shihezi Formation of the Permian Lower Series, the Upper Shihezi Formation of the Permian Upper Series, and Quaternary loess. The comprehensive column section of strata from the surface of the coal mining layer is shown in Figure 2.

| System | Formation | Member | Buried Depth | Columnar | Thickness | Lithology | Remarks |
|---|---|---|---|---|---|---|---|
| Quaternary | | | 40.2m | | 40.2m | Loess | |
| Permian | Upper Shihezi Formation | Upper Member | 62.9m | | 22.7m | Sandstone | |
| | | | 107.1m | | 44.2m | Sandy Mudstone | |
| | | | 172.3m | | 65.2m | Mudstone | |
| | | | 219m | | 46.7m | Sandstone | Main key stratum |
| | | Middle Member | 240.6m | | 21.6m | Sandstone | Inferior key stratum 1 |
| | | | 256.9m | | 16.3m | Sandy mudstone | |
| | | | 271.6m | | 14.7m | Mudstone | |
| | | | 283.7m | | 12.1m | Sandy Mudstone | |
| | | | 306.1m | | 22.4m | Sandstone | Inferior key stratum 2 |
| | | Lower Member | 350.6m | | 44.5m | Sandy Mudstone | |
| | | | 388m | | 37.4m | | |
| | | | 448.6m | | 60.6m | Sandstone | |
| | | | 462.3m | | 13.7m | Mudstone | |
| | | | 473.1m | | 10.8m | Sandstone | |
| | Lower Shihezi Formation | | 493.3m | | 20.2m | Mudstone | |
| | | | 519.2m | | 25.9m | Sandstone | Inferior key stratum 3 |
| | | | 537.8m | | 18.6m | Mudstone | |
| | | | 547.6m | | 9.8m | Sandy Mudstone | |
| | Shanxi Formation | | 604.3m | | 56.7m | Sandstone | |
| | | | 609.8m | | 5.5m | Coal | |

**Figure 2.** Comprehensive column section.

The working face is the working site for the direct mining of minerals or rocks. In coal mines, the working face is the first site of coal production. The overburden bed separation grouting project was carried out at the 8006 working face of Wuyang Coal Mine. This working face is 280 m wide and 1140 m long. The main mining seam is No. 3 coal, with a burial depth of approximately 600 m and an average coal seam thickness of 5.5 m. The injected bed separation space is located below the inferior key stratum 2 at the boundary between the middle and lower part of the Upper Shihezi Formation. The burial depth of the injected bed separation space is about 300 m. The grouting material is a mixture of mine water and fly ash. The fly ash slurry produced by the grouting station is pumped and then piped to the grouting borehole in the working face for grouting.

## 3. Methodology

The methodology mainly includes 3D seismic interpretation of faults in the study area and the numerical simulation of the slurry transport processes. First, 3D seismic exploration is carried out in the study area, and the 3D seismic data are finely interpreted to obtain the distribution of faults in the study area. Afterwards, the slurry leakage channels of the bed separation slurry were obtained through theoretical analysis based on the spatial relationship among the bed separation, faults and surface grout leakage points at the project site. Finally, numerical simulation was used to simulate the slurry transport process.

### 3.1. 3D Seismic Interpretation

The basis of 3D seismic exploration is the theory of seismic wave propagation in the medium. Artificially created seismic waves are reflected and refracted on different subsurface lithological dividing surfaces and returned to the ground, causing ground vibrations, and using ground seismometers to collect seismic records. Finally, the seismic information is used to interpret the stratigraphic and geological structure information to achieve the purpose of exploration. The seismic area is located southwest of the Wuyang Mine, with a length of 3000 m in the east–west direction and 1300~1946 m in the north–south direction. The 8006 working face is located in the southeast of the survey area. Nine boreholes near the survey area were selected for the study. The extent of the 3D seismic area and the location of the boreholes are shown in Figure 3. The borehole data and logging curve data were preprocessed with the 3D seismic data and the surrounding geological data in the study area, and were used to study the characteristics of stratigraphic development, tectonic relief patterns, fault development periods, and tectonic evolution processes.

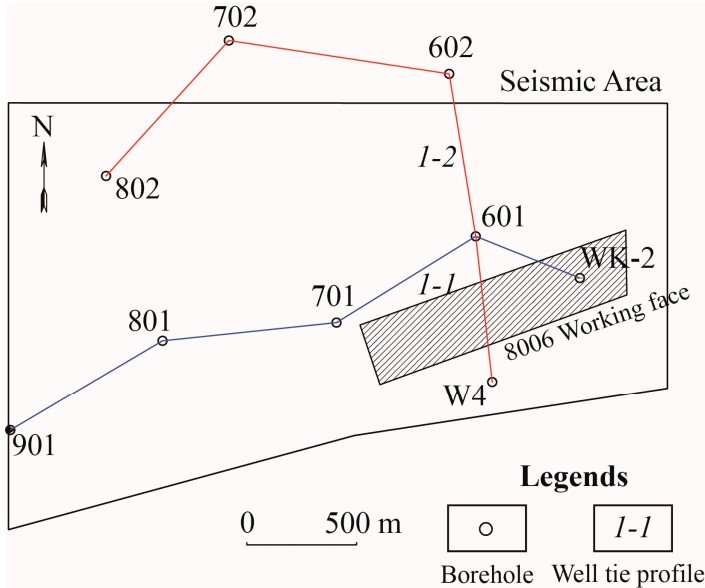

**Figure 3.** Seismic area.

The information from the boreholes in the area reveals that the stratigraphic section from the top of the Ordovician to the Quaternary is stable and well developed, the curves of the stratigraphic interface are clearly marked, and the comparison between wells is intuitive. The nine boreholes were made into two well tie profiles (Figure 4), and the comparison of the logging curves of the main boreholes in the area shows that the characteristics of the logging curves are highly similar, the thickness of the stratigraphy changes continuously, and the comparison between wells is reasonable for geological stratification.

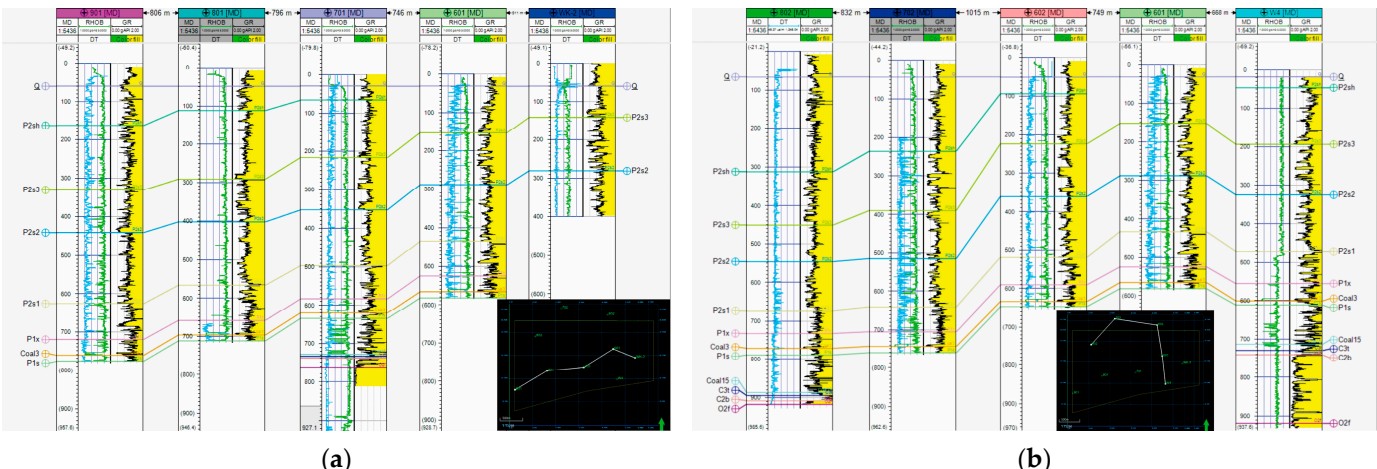

**Figure 4.** Well tie profiles: (**a**) 1-1 well tie profiles; (**b**) 1-2 well tie profiles.

Seismic horizon calibration is conducted by using a synthetic seismogram. The No. 3 coal seam quarried at the 8006 working face is of large thickness and has a large difference in geophysical properties with the top and bottom seams, forming a very prominent and continuous seismic reflection feature in the middle of the seismic profile in the whole area, which is the main reference mark when calibrating the synthetic seismogram. The geophysical properties of the loose sediments of the Quaternary differ significantly from those of the underlying Permian strata, creating a large wave impedance difference at the bottom boundary of the Quaternary that makes the seismic reflections distinctive. The Quaternary can also be used as a reference marker for the calibration of synthetic seismograms. Using the two important geological interfaces mentioned above as a reference, the synthetic seismograms of the boreholes in the seismic area were calibrated. The results show that the synthetic seismic records of all boreholes correspond well with the seismic profiles. The 701 borehole synthetic seismic records are shown in Figure 5. The seismic profile of the continuous well shows that the geological stratification and the seismic homogeneous axis correspond accurately, indicating that the geological stratification is reasonable. The seismic profile of the continuous well is shown in Figure 6.

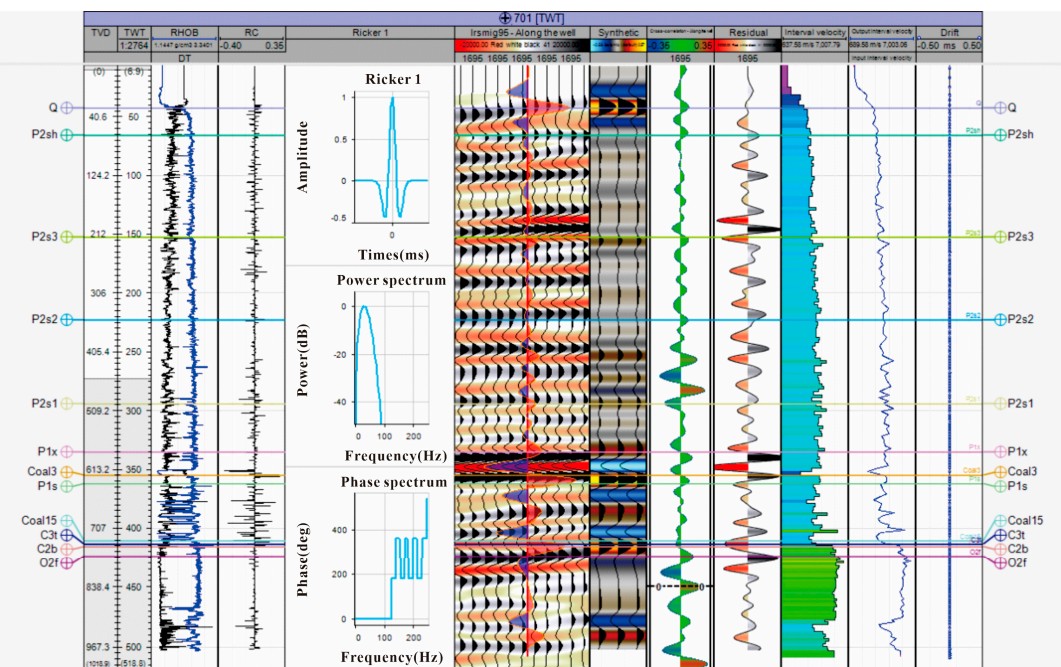

**Figure 5.** Synthetic seismogram of borehole 701.

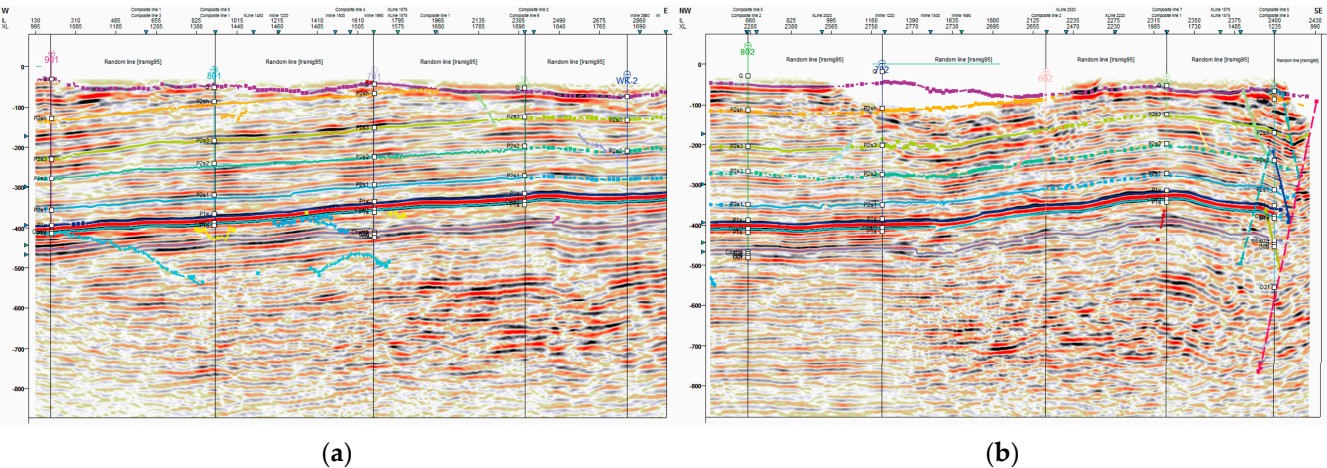

**Figure 6.** Well tie seismic profiles: (**a**) 1−1 well tie seismic profile; (**b**) 1−2 well tie seismic profile.

The completion of 3D stratum horizon interpretation is a prerequisite for fault interpretation. In the process of 3D seismic stratigraphic interpretation in the study area, first, the seismic stratigraphic position is determined through the well tie profile, and the stratigraphic profile of the continuous well line is interpreted. Second, a combination of arbitrary lines and longitudinal and transverse lines are used to compare and trace the stratigraphic positions according to the relationship between the wave groups and the waveform characteristics of each reflection layer in the study area to ensure the uniformity of the stratigraphic positions in the study area. Finally, in some profiles with low signal-to-noise ratios and no continuous homogeneous axes, inferential interpretations are made based on the thickness of the upper and lower reflector layers, tectonic change trends, and geological data to ensure that the interpretation scheme is consistent with the regional geological and tectonic model.

Time–depth conversion is the process of converting a time-domain plane map to a depth-domain plane map, and is the inverse operation of velocity calibration. By multiplying the velocity of each point on the time domain plan by half of the reflection time at that point, a depth domain plan can be derived. Using the bottom boundary of the No. 3 coal seam as an example, the time domain structure map and depth domain structure map are shown in Figure 7.

The 3D seismic body is viewed to understand the general tectonic features within the seismic area and to make full use of the horizontal slicing and profiling techniques of 3D seismic to interpret faults. On the seismic profiles, distortions, bifurcations, mergers and occurrences in the homogeneous axis are identified, and the fault development's precise location is determined. A range of technical methods are used to intervalidate the interpretation to ensure its accuracy. The plane combination of faults was determined using temporal slices and variance body slices along the layers. The consistency of fault planes and sections was verified using a combined display of horizontal slices and profiles. The reasonableness of the interpretation of the faults and the spatial combination of the faults is tested by drawing arbitrary orientation profiles. The layer leveling sections are used to verify the tectonic morphology, the location of tectonic highs, and intertectonic relationships and the evolution of the tectonics in the longitudinal direction, as well as the relationship among the faults. Finally, the fault interpretation results are obtained. After the completion of the stratigraphic and fault interpretation, a quality check of the stratigraphic and fault interpretation results was carried out using a full 3D display to achieve a closed and uniform interpretation result for the whole area.

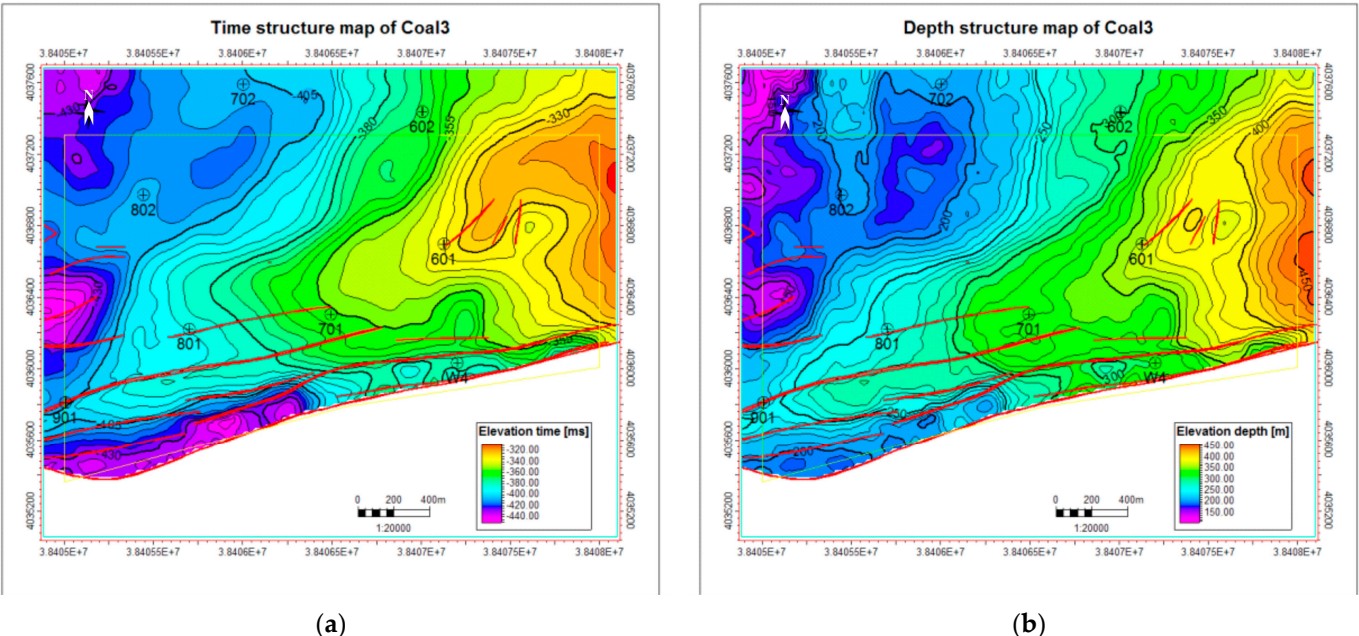

**Figure 7.** No. 3 Coal seam time-domain plane map and time-domain plane map: (**a**) Time-domain plane map; (**b**) Time-domain plane map.

*3.2. Establishment of Slurry Transport Numerical Model*

In this section, COMSOL Multiphysics is used to simulate the transport of the slurry from the injected bed separation to the surface and the leakage at the surface. The more established theory of slurry diffusion is the cubic law, which estimates the viscous flow between smooth parallel plates. However, many scholars have confirmed that the cubic law does not apply in contexts with high roughness or large aperture when they can be addressed by the Reynolds lubrication equation or the Navier–Stokes equation [34–36]. In the coal mining process, there is a dynamic development of the bed separation space as the overburden moves. According to the field drilling detection results of the overburden bed separation grouting project at the 8006 working face, the maximum height of the injected bed separation can reach approximately 1 m, which is beyond the scope of the application of the cubic law. Therefore, according to the size of the bed separation and its corresponding slurry diffusion model, the numerical simulation is divided into two cases: small bed separation and large bed separation. When the bed separation is relatively small, bed separation can be regarded as the space between smooth parallel plates, and slurry diffusion satisfies the cubic law. The numerical model can be calculated using Darcy's law method of the groundwater percolation module. When the bed separation is relatively large, the diffusion of the slurry does not satisfy the cubic law, and the Navier–Stokes equation of the computational fluid dynamics module is used to calculate it.

3.2.1. Numerical Model for the Transport of Slurry Injected into Small Bed Separation

A three-dimensional numerical model was developed using the finite element numerical simulation software COMSOL Multiphysics. The model is 1540 m long and 846 m wide and has an average height of approximately 800 m. The stratification of the model was first generalized based on the geological reports and borehole data of the study area. The model was divided into 10 layers. Layer 1 represents the Quaternary loess, layer 2 is the sandstone of the Shiqianfeng Formation, layer 3 is the mudstone of the upper section of the Upper Shihezi Formation, layer 4 is the sandstone of the middle section of the Upper Shihezi Formation, layers 5 and 6 are the mudstone and sandstone of the lower section of the Upper Shihezi Formation, layers 7 and 8 are the mudstone and sandstone of the Lower Shihezi Formation, layer 9 is the coal seam, and layer 10 is the coal seam floor. Corresponding to the location of the injected bed separation in the field, a small bed separation was set

between layers 4 and 5 of the model. The small bed separation was set in the middle of the working face width direction, with the projection of the starting boundary on the surface coinciding with the starting boundary of the working face. The length and width of the small bed separation are both 140 m, and the grouting hole is located in the centre of the bed separation. Details of the numerical model are shown in Figure 8.

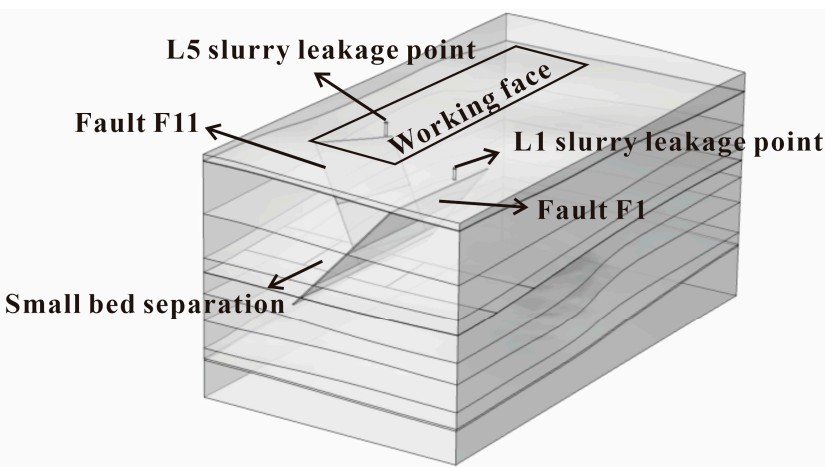

**Figure 8.** Numerical model of small bed separation slurry transport.

The model is calculated by the two-phase Darcy's law. The model assumes that the separation position will not change after the separation space is filled with grout; the slurry is an incompressible homogeneous isotropic fluid; the model medium is isotropic and incompressible, and the permeability remains constant; and the coexistence space of groundwater and slurry is regarded as Darcy's law two-phase seepage field. After simplification, the seepage direction of groundwater remains uniform, and the pressure is constant. The control equation of the model is composed of a dynamic equation, mass conservation equation, and fracture seepage equation, as shown in Equations (1)–(6).

The mass conservation equation of slurry is shown in Equation (1). The mass conservation equation of water is shown in Equation (2).

$$-\left[\frac{\partial(\rho_s v_{sx})}{\partial x} + \frac{\partial(\rho_s v_{sy})}{\partial y} + \frac{\partial(\rho_s v_{sz})}{\partial z}\right] = \frac{\partial(\varepsilon_p \rho_s s_s)}{\partial t} \tag{1}$$

$$-\left[\frac{\partial(\rho_w v_{wx})}{\partial x} + \frac{\partial(\rho_w v_{wy})}{\partial y} + \frac{\partial(\rho_w v_{wz})}{\partial z}\right] = \frac{\partial(\varepsilon_p \rho_w s_w)}{\partial t} \tag{2}$$

where $\rho_s$ is the slurry density, kg/m$^3$; $s_s$ is the volume fraction of slurry in the medium; $\rho_w$ is the density of water, kg/m$^3$; $t$ is the grouting time; $v$ is the seepage field velocity, m/s; vs. is the slurry flow velocity, m/s; $v_w$ is the water flow velocity, m/s; $s_w$ is the volume fraction of water in the medium; $\mu$ is the fluid viscosity, Pa·s; $K$ is the permeability of the medium, m$^2$; $\varepsilon_p$ is the porosity of the medium; $c_w$ is the mass fraction of water in the medium; $p$ is the slurry pressure, MPa; and $D_c$ is the capillary diffusion coefficient, m$^2$/s.

Equations (3)–(5) are dynamic equations, also known as the generalised Darcy's law, and the control equation is Equation (6).

$$v = \frac{K}{\mu}\Delta p \tag{3}$$

$$v_s = s_s v \tag{4}$$

$$v_w = s_w v \tag{5}$$

$$s_s + s_w = 1 \tag{6}$$

The model inlet boundary conditions are pressure boundary conditions or velocity boundary conditions, and the rest of the boundaries are no-flux boundaries, meeting the no-slip conditions. The initial condition of the volume fraction of water is 1. 1 cm, and is taken as the height of the separation in the small separation model, according to the actual engineering and numerical simulation experience. The rest of the model parameters are selected reasonably; see Table 1 for details.

**Table 1.** Parameters for the small bed separation numerical model.

| Parameters | Permeability (m$^2$) | | | Grouting Pressure (MPa) | Grouting Hole Diameter (mm) | Slurry Viscosity (Pa·s) | Grouting Speed (m/s) |
|---|---|---|---|---|---|---|---|
| | Sandstone | Mudstone | Fault | | | | |
| Value | $1 \times 10^{-16}$ | $1 \times 10^{-18}$ | $1 \times 10^{-13}$ | 3 | 150 | 0.0025 | 2.00 |

3.2.2. Numerical Model for the Transport of Slurry Injected into Large Bed Separation

The size and stratification of the large bed separation slurry transport model are consistent with the small separation layer model, but the size of bed separation space is different. In this model, the separation layer is 280 m in length and width, which is also set between the fourth and fifth layers of the model. The projection of the three boundaries of the bed separation on the surface coincides with the working surface boundary, and the grouting hole is located in the centre of the large bed separation. The model is shown in Figure 9.

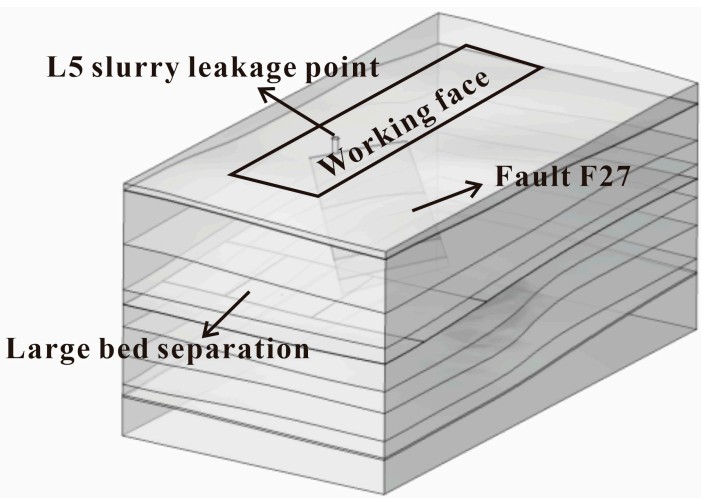

**Figure 9.** Numerical model of large bed separation slurry transport.

The numerical model with large bed separation is calculated by the method of multi-physics field coupling of phase field and the two-phase Darcy's. The assumptions of the model are consistent with those of the numerical model with small bed separation. The diffusion of slurry in the large bed separation is simulated by the phase-field method, and the diffusion of slurry in the fault is simulated by the two-phase Darcy's law. The diffusion of slurry in the ionosphere meets the control Equations (1)–(6), and the diffusion of slurry in the fault meets the control Equations (7) and (8).

The equation of motion is shown in Equation (7).

$$\rho \frac{\partial \vec{u}}{\partial t} + \rho \left( \vec{u} \nabla \right) \vec{u} = -\nabla p + \nabla \left\{ \mu \left[ \nabla \vec{u} + \left( \nabla \vec{u} \right)^T \right] \right\} + \vec{F} \tag{7}$$

where $\rho$ is the slurry density, kg/m$^3$; $\vec{u}$ is the velocity field; $p$ is the slurry pressure, MPa; $\mu$ is the slurry dynamic viscosity, Pa·s; $\vec{F}$ is the external force acting on the slurry, N. Equation (7) is the full description of the Navier–Stokes equation.

The continuity equation is shown in Equation (8).

$$\nabla \vec{u} = 0 \tag{8}$$

The continuity equation represents the conservation of mass and the invariance of volume in the control area during the movement process.

The inlet boundary condition of the model is the pressure boundary condition or velocity boundary condition, the boundary of the large bed separation is a no-slip boundary, and the rest of the boundary is a no-flux boundary; they all satisfy the no-slip condition. The initial condition is that the volume fraction of water is 1. The height of the bed separation in the large bed separation model is taken as 1 m. According to the actual engineering and numerical simulation experience, the remaining parameters of the model are reasonably selected. See Table 2 for details.

**Table 2.** Parameters for the large bed separation numerical model.

| Parameters | Permeability (m²) | | | Grouting Pressure (MPa) | Grouting Hole Diameter (mm) | Slurry Viscosity (Pa·s) | Grouting Speed (m/s) |
|---|---|---|---|---|---|---|---|
| | Sandstone | Mudstone | Fault | | | | |
| Value | $1 \times 10^{-16}$ | $1 \times 10^{-18}$ | $1 \times 10^{-13}$ | 3 | 150 | 0.0025 | 0.006 |

## 4. Results and Discussion

### 4.1. Results of Seismic Interpretation

After the detailed interpretation of the 3D seismic data, 46 faults are obtained in the survey area, including 45 normal faults and one reverse fault. There are 15 normal faults developed near the 8006 working face, which are F1, F10, F11, F17, F18, F19, F20, F21, F22, F23, F25, F26, F27, F28, and F38. The distribution of faults near the working face is shown in Figure 10.

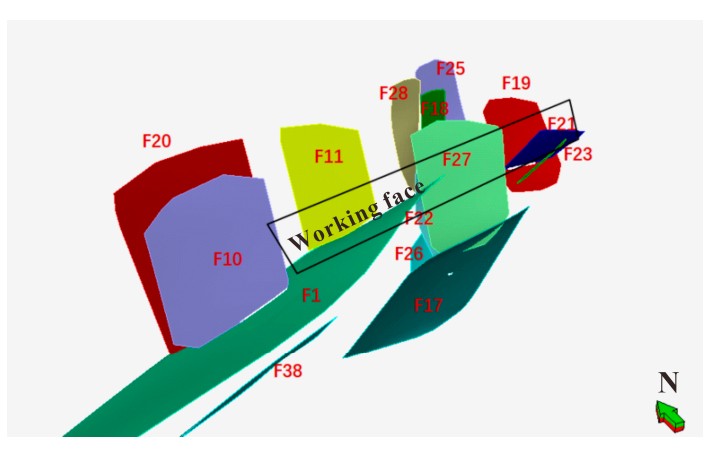

(**a**)

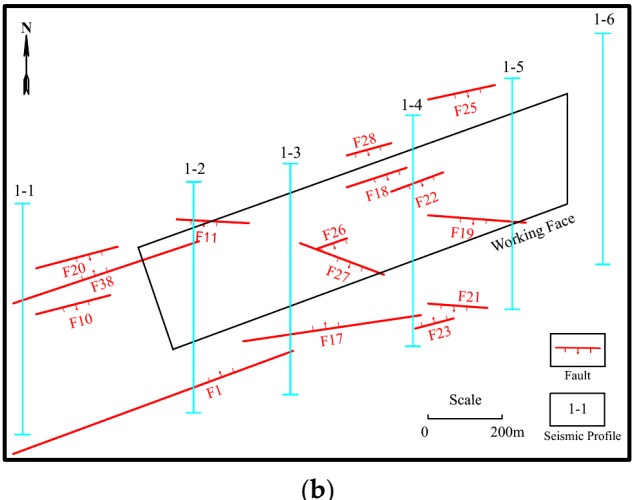

(**b**)

**Figure 10.** Distribution of faults near the working face: (**a**) Three-dimensional distribution of faults; (**b**) Fault plane distribution and location of seismic profiles.

Six typical seismic profiles near the working face are shown in Figure 11, and the locations of the profile lines are shown in Figure 10b. The faults extend vertically far upwards as a whole, developing from the Shanxi Formation through the Quaternary. The faults are mainly developed in the axis and the wing of anticlines. The faults are poorly closed, as they are all tensor positive faults. The faults are also interconnected, which can easily constitute a three-dimensional channel for slurry leakage. In summary, the complex structure of faults near the working face has an impact on the transport of slurry for bed

separation injection. Among them, the five faults of F1, F11, F18, F19 and F27 penetrate the surface and are important channels for slurry diffusion to the surface.

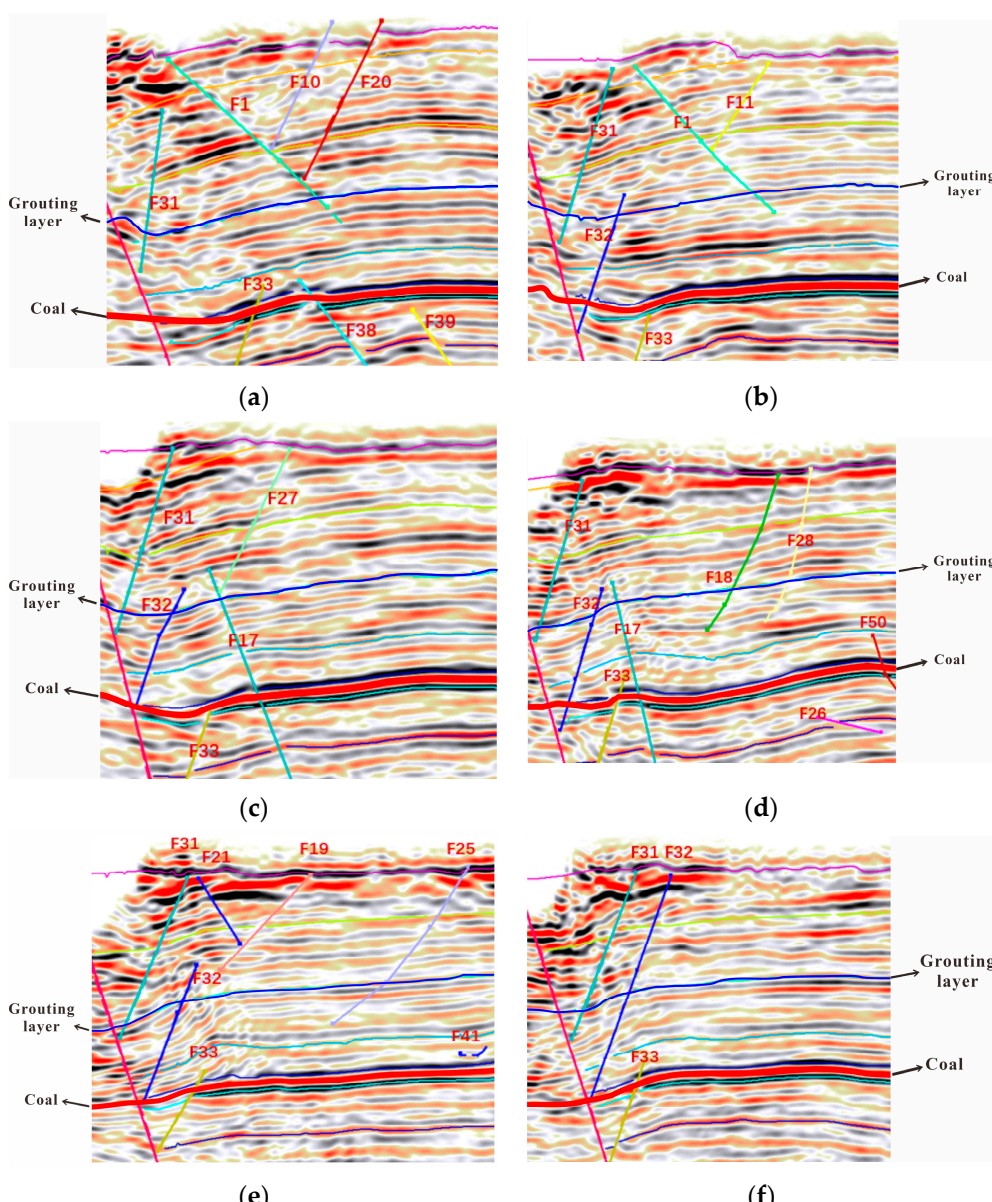

**Figure 11.** Profiles of seismic interpretation results: (**a**) Profile 1-1; (**b**) Profile 1-2; (**c**) Profile 1-3; (**d**) Profile 1-4; (**e**) Profile 1-5; (**f**) Profile 1-6.

### 4.2. Slurry Leakage Channel Analysis

The permeability of the bed separation space and fault are better than the permeability of the surrounding rock. The faults near the working face are developed and extend far vertically. Some of the faults are connected with the injected bed separation, forming fault channels for slurry diffusion. When the slurry diffusing along the bed separation touches the upper wall of the bed separation space, the slurry almost fills the bed separation space, and at this time, the continuously injected slurry will diffuse to the faults with better permeability under the action of grouting pressure. If the fault extends upwards to the surface, the slurry will diffuse along the fault to the surface and form a surface grout leakage. The development and extension of each fault in the vertical direction are shown in the seismic interpretation result profiles shown in Figure 11. There are eight slurry leakage

points at the site of the bed separation grouting project, and the locations of slurry leaks and the plane distribution of related faults are shown in Figure 12.

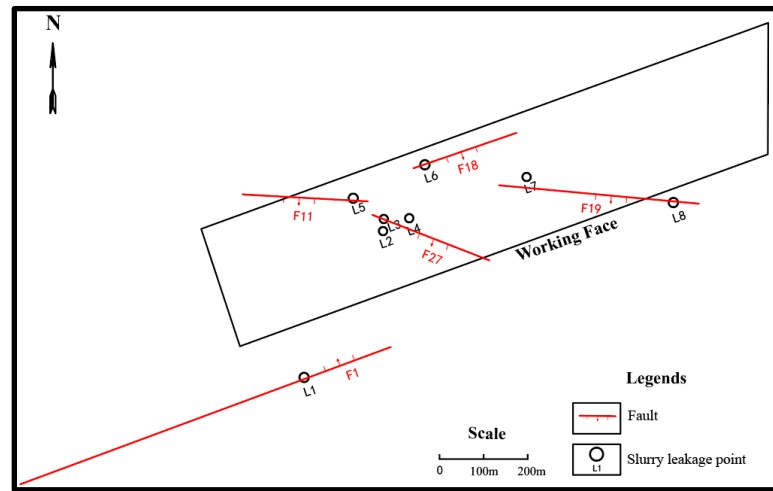

**Figure 12.** Plan view of the location of slurry leakage points and faults.

From Figure 12, L1 is located near the F1 fault. L2, L3 and L4 are located near the F27 fault. L5 is located near the F11 fault. L6 is located near the F18 fault. L7 and L8 are located near the F19 fault. The injected bed separation space of this overburden bed separation grouting project is located at the boundary between the middle and lower sections of the Upper Shihezi Formation, with an average buried depth of 300 m. The starting depths of the F1 fault, the F27 fault, the F18 fault, and the F19 fault are 470 m, 350 m, 450 m and 320 m, respectively, and all four faults are connected with the injected bed separation and extend vertically to the surface.

After the bed separation grouting project began, the slurry transported successively to the surface through the bed separation and fault channels under the action of grouting pressure, causing slurry leakage at L1, L3, L6, and L8. The field investigation shows that bedrock weathering fissures are developed in the study area. When the slurry passes through the bed separation into the fault and transports along the F27 and F19 faults near the surface, it then transports along the bedrock weathering fissures to the surface, causing slurry leakage at L4, L5, and L7. It can be seen from the seismic profile shown in Figure 11b that the F11 fault is connected with the F1 fault at its starting position and extends upwards to the surface. During the slurry injection process, the slurry transported successively to the surface through the bed separation, F1 fault and F11 fault channels, causing slurry leakage at L5. Photos of the slurry leakage site are shown in Figure 13. To sum up, combined with the geological structure characteristics of the study area and the spatial location relationship between the injected bed separation, faults and slurry leakage points, it was determined that the main running channels of the slurry were bed separation and faults.

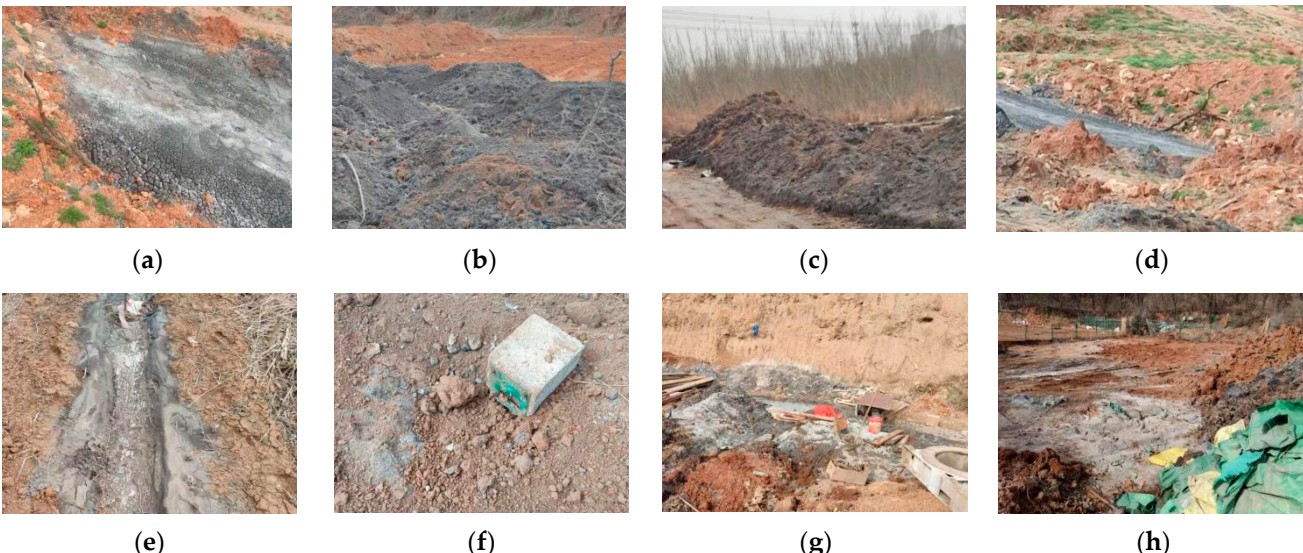

**Figure 13.** Slurry leakage site photos: (**a**) L1 slurry leakage point; (**b**) L2 slurry leakage point; (**c**) L3 slurry leakage point; (**d**) L4 slurry leakage point; (**e**) L5 slurry leakage point; (**f**) L6 slurry leakage point; (**g**) L7 slurry leakage point; (**h**) L8 slurry leakage point.

*4.3. Numerical Simulation Results of the Slurry Transport Process*

According to the previous analysis, the faults are important channels for the longitudinal transport of slurry from the injected bed separation space to the surface. To simulate the slurry transport process, faults are added to the small and large bed separation slurry transport models, and numerical simulations are performed for a typical slurry leakage case in the overburden bed separation grouting project at the 8006 working face of the Wuyang coal mine.

4.3.1. Numerical Simulation Results for Small Bed Separation-Fault Slurry Transport

During the initial grouting of the 8006 working face, slurry runs occurred at L1 and L5. According to the previous analysis, the F1 fault connects the separation space and the surface and it is the independent channel for slurry leakage from the injected bed separation space to the surface at L1. The beginning of the F11 fault is connected to the F1 fault. The slurry transported through the injected bed separation space, F1 fault and F11 fault to the surface, causing slurry leakage at L5. L1 and L5 represent two modes of slurry leakage through independent fault channels and combined fault channels, respectively. Therefore, the L1 and L5 slurry leakages are used as typical slurry leakage cases for numerical simulation studies. Two faults, F1 and F11, are added to the model to simulate the slurry leakage process at L1 and L5.

The range of slurry transport at different times obtained from the numerical simulation is shown in Figure 14. From the calculation results, it can be seen that after the start of slurry injection, the slurry first transported in the injected bed separation space. When the slurry transported to the position where fault F1 penetrates the bed separation, part of the slurry starts to enter fault F1 and travels vertically along fault F1. When the slurry transported along fault F1 to the position where fault F11 penetrates fault F1, some of the slurry enters fault F11 from fault F1. After this time, the slurry transported into both the bed separation space and the two faults. Eventually, 10 days after the start of grouting, the slurry transported to the surface along the F1 fault, and slurry leakage occurred at L1. Thirteen days after the start of injection, the slurry transported to the surface along the F11 fault, resulting in slurry leakage at L5.

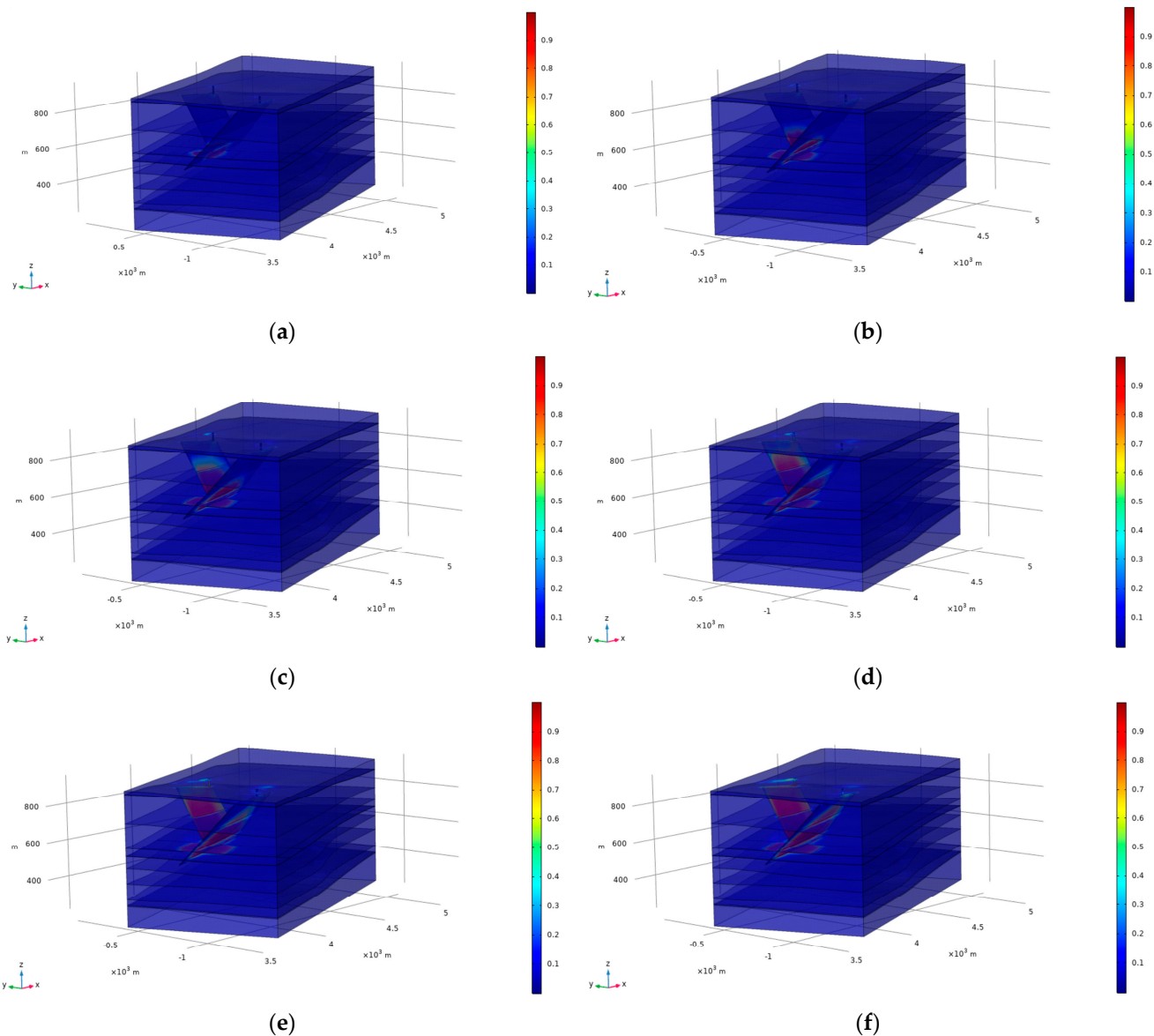

**Figure 14.** Slurry transport clouds at different times: (**a**) 1d; (**b**) 3d; (**c**) 5d; (**d**) 7d; (**e**) 10d; (**f**) 13d.

From the simulation results, it can be seen that it took 10 days to transport the slurry to the surface at L5 and 13 days to transport it to L1, while in the actual field project, slurry leakage occurred at L5 and L1, respectively, 10 days and 14 days after the start of grouting. The actual time of surface leakage is close to the calculated results. It can be seen that the simulation results of the slurry transport process have high reliability. The numerical simulation results provide validation for the conclusions of the previous theoretical analysis, confirming that bed separation and faults are the main channels for slurry leakage.

### 4.3.2. Numerical Simulation Results for Large Bed Separation-Fault Slurry Transport

Slurry leakage occurred at L3 during grout injection into bed separation on site. According to the previous analysis, the F27 fault is connected to the injected bed separation at the beginning and extends upwards to the surface. After the slurry was injected into the bed separation under the pressure of the grouting, it transported through the bed separation and the F27 fault to the surface in turn, resulting in a slurry leakage at L3. The slurry leakage at L3 is taken as a typical case for numerical simulation studies, so the F27 fault is added to the model to simulate the slurry leakage process at L3.

The range of slurry transport at different times obtained from the numerical simulation is shown in Figure 15. From the numerical simulation results, it can be seen that after the start of slurry injection in the bed separation, the slurry first transports in the injected bed separation. When it reaches the position where the F27 fault penetrates the injected bed separation, part of the slurry starts to enter the F27 fault and travels vertically along the F27 fault. Eventually, 12 days after the start of grouting, the slurry transported to the surface along the F27 fault, and slurry leakage occurred at L3.

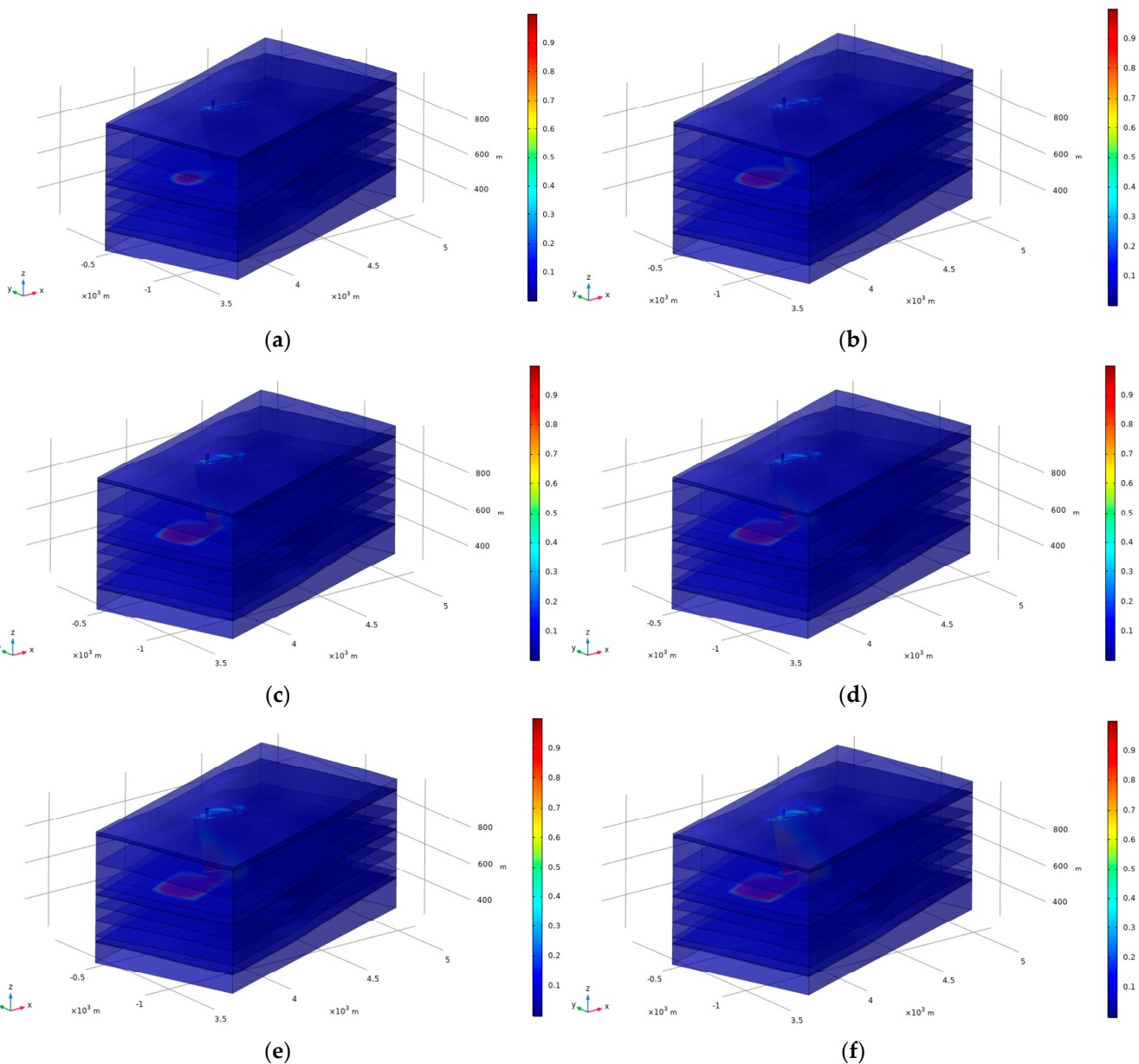

**Figure 15.** Slurry transport clouds at different times: (**a**) 1d; (**b**) 3d; (**c**) 5d; (**d**) 7d; (**e**) 10d; (**f**) 13d.

The numerical simulation results show that it took 12 days for the slurry to transport at location L3. At the actual grouting site, a slurry leakage occurred at location L3 twelve days after the start of grouting, and the actual slurry leakage occurred at the same time as the calculated results. This shows that the numerical simulation results of the slurry transport process have high reliability. The numerical simulation results provide validation for the conclusions of the previous theoretical analysis, confirming that bed separation and faults are the main channels for slurry leakage.

In summary, typical slurry leakage cases are selected for numerical simulation under the conditions of a small bed separation space and a large bed separation space. The slurry leakage times obtained from the numerical simulations were consistent with the actual slurry leakage times. The results of both simulations show that the slurry is first transported in the injected bed separation. The slurry will enter into the fault at the junction of the bed separation space and the fault, and it will enter into another fault at the junction of the two faults, but it will hardly be transported in the surrounding rock. The numerical simulation results provide good evidence for the separation channel and fault channel of slurry leakage obtained from theoretical analysis.

The research in this paper proves that the channels for grout leakage to the surface of an overburden separation grouting project are the injected bed separation space and the faults. It can be seen that the distribution of faults in the vicinity of the grouting working face should be given due attention by engineers in overburden bed separation grouting projects. To reduce the blindness of the grouting project, the distribution of faults in the vicinity of the grouting working face should be explored before grouting begins. Large faults penetrating the injected bed separation space and the surface can be sealed if necessary in order to prevent surface slurry leakage.

## 5. Conclusions

In this paper, the distribution of faults near the 8006 working face of Wuyang Coal Mine has been proven by using 3D seismic exploration technology. On this basis, according to the spatial position relationship of the injected bed separation, faults and surface slurry leakage point, the slurry leakage channels of each slurry leakage point are theoretically analysed. The transport process of grout from the injected bed separation space to the surface is then simulated by a numerical simulation method. The following conclusions are drawn:

(1) A total of 46 faults were detected in the seismic exploration area, including 45 normal faults and one reverse fault. There were 15 normal faults developed near the 8006 working face, which were the F1, F10, F11, F17, F18, F19, F20, F21, F22, F23, F25, F26, F27, F28, and F38 faults.

(2) Combined with the geological structure characteristics of the study area and the spatial location relationship among the injected bed separation, faults and field grout leakage points, based on the difference in permeability among the surrounding rock, bed separation and faults, it is proposed that the main grout leakage channels of overburden bed separation grouting are the bed separation and the fault. The five faults: F1, F11, F18, F19, and F27 near the working face connect to the surface and are important channels for slurry leakage to the surface.

(3) The numerical simulation results of the slurry transport process show that the slurry is mainly transported in the injected bed separation and the faults connecting with the bed separation. In the small separation layer fault model, after grouting starts, the grout flows into bed separation, the F1 fault and the F11 fault, successively, for transport. Finally, the slurry leaks. In the large bed separation model, after grouting starts, the grout flows into the bed separation and F27 fault in turn for transport. Finally, the slurry leaks. The results of numerical simulation show that the main slurry transport channels are the injected bed separation space and faults.

(4) As the permeability of bed separation and faults is far greater than that of the surrounding rock, the slurry will preferentially transport in the space of injected bed separation. When the fault is connected with the bed separation space, the slurry transported to the junction will enter the fault and migrate along the fault. If the fault extends upwards to the surface, the slurry may continue to migrate to the surface, causing leakage of the surface slurry.

In this paper, the slurry leakage channel in an overburden bed separation grouting project has been determined, and the slurry transport process has been simulated. The research results have important guiding significance for the layout of grouting holes, the

determination of the distribution of slurry and its solid structure, the control of grouting volume and the prevention of surface slurry leakage in overburden bed separation grouting projects. However, there are still some problems that require further study. In this paper, the dynamic development process of the injected bed separation space is not considered when the slurry transport process is simulated. In subsequent research, new numerical simulation methods or model test methods can be used to reflect the actual dynamic development process of bed separation in the grouting process, and on this basis, research on the slurry transport process can be carried out to continue to improve the slurry transport theory of overburden bed separation grouting projects.

**Author Contributions:** Conceptualization, S.W., W.M. and N.X.; methodology, T.K.; validation, Z.G., Z.L. and Y.H.; formal analysis, T.K., W.Y. and W.M.; investigation, T.K., Z.G. and S.W.; resources, Z.L. and Y.H.; writing—original draft preparation, T.K.; writing—review and editing, W.M. and H.L. All authors have read and agreed to the published version of the manuscript.

**Funding:** This research was funded by the Fundamental Research Funds for the Central Universities, grant number 2-9-2020-019.

**Data Availability Statement:** Not applicable.

**Acknowledgments:** Thanks to Yanfang Ye for her suggestions on English writing. The authors would like to thank the editor and the reviewers for their constructive suggestions.

**Conflicts of Interest:** The authors declare that they have no conflict of interest.

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
