# Peer review of "Slurry Leakage Channel Detection and Slurry Transport Process Simulation for Overburden Bed Separation Grouting Project: A Case Study from the Wuyang Coal Mine, Northern China"

_water, doi:10.3390/w15050996_

Round 1
Reviewer 1 Report (Previous Reviewer 1)
The topic of the manuscript entitled “Slurry Leakage Channel Detection and Slurry Transport Process Simulation for Overburden bed separation grouting project :a Case Study from the Wuyang Coal Mine, Northern China” is interesting. In this paper, the spatial distribution of faults in the study area was detected. Based on the spatial location of the injected bed separation, faults and surface slurry leakage points, the specific slurry leakage channels and the slurry leakage process of the eight slurry leakage points were analyzed. Finally, the slurry transport process was simulated using a two-phase flow numerical model and the simulation results provide good support for the theoretical analysis results. The conclusions of this manuscript are of great guidance for grout engineering design and environmental protection of overburden bed separation grouting projects. The text is well organized and the figures and tables in the manuscript are of high quality.
In resubmitted revised manuscript, the authors have extensively revised the manuscript in response to previous reviewers' comments. A detailed response to the previous reviewers’ comments was provided, so the manuscript is almost ready to publish in my opinion. However, there is a minor error in the manuscript. In the annotation of the title in Figure 13, “L1 slurry leakage points” should be changed to “L1 slurry leakage point” and so on.
I recommend accepting this manuscript for publication in Water after completing the above revisions.
Author Response
Point 1: In the annotation of the title in Figure 13, “L1 slurry leakage points” should be changed to “L1 slurry leakage point” and so on.
Response 1: Thanks for the reviewer's suggestion. We revised the manuscript (Lines 387-389).

Reviewer 2 Report (New Reviewer)
Please would you like to presents the results with the subject late portion in the Section 1 Introduction.
Figure 4, 5 and 6 are poor in quality for illustration, and revise to high quality in order to be east taking understandings.
Author Response
Point 1: Please would you like to presents the results with the subject late portion in the Section 1 Introduction.
Response 1: Thanks for the reviewer's suggestion. As suggested by the reviewers, we present the results of this paper in Section 1 Introduction(Lines 99-111).
Point 2: Figure 4, 5 and 6 are poor in quality for illustration, and revise to high quality in order to be easy taking understandings.
Response 2: Thanks for the reviewer's suggestion. Figures 4, 5 and 6 are used to show the results of certain steps in the seismic interpretation process and are not intended to be used as illustrations. We have revised Figures 4, 5 and 6 and replaced them with clearer versions in the manuscript(Lines 177, 194 and 208).

Reviewer 3 Report (New Reviewer)
The papers is well written and the topic is interested. I suggest the paper for publication in its current form.
Author Response
Point 1: The papers is well written and the topic is interested. I suggest the paper for publication in its current form.
Response 1: The authors are very grateful to the reviewers for their approval.

This manuscript is a resubmission of an earlier submission. The following is a list of the peer review reports and author responses from that submission.
Round 1
Reviewer 1 Report
In coal mine overburden bed separation grouting projects, detecting slurry leakage channels and determining the slurry leakage process are prerequisites for slurry leakage prevention. Therefore, the topic of this paper is interesting In this paper, the spatial distribution of faults in the study area was detected. Based on the spatial location of the injected bed separation, faults and surface slurry leakage points, the specific slurry leakage channels and the slurry leakage process of the eight runout sites were analysed. Finally, the slurry leakage process is simulated and the simulation results provide good support for the theoretical analysis results. The results of the study are presented as a guide for slurry run prevention and control. The paper is well written, but there are still some parts that need to be further modified as specified below.
1. How does this study compare with previous studies on the flow and transport of slurry from overburden bed separation grouting projects? This question needs to be further clarified in the introduction.
2. The location of the seismic area is given directly in Figure 1, while the overall location of the Wuyang Mine is unclear.
3. The term grouting layer is inaccurate in the remarks to Figure 2. The grouting should be carried out in the bed separation space below the layer. It is suggested that this statement be amended.
4. In the section 3.2, two simulations are carried out for large and small bed separation as the cubic law has its own range of applicability on the scale. However, only one reference is cited in this paper to describe the range of applicability of the cubic law, and it is recommended that additional references be added to support this theory.
5. It is suggested that a clearer indication of the location of the injected bed separation space and coal seam be added to Figure 11.
6. How do each of the eight photographs in Figure 12 correspond to the slurry leak points in Figure 11? Write them out at the figure names.
I suggest a minor revision.
Reviewer 2 Report
The manuscript could be of some interest for researchers and technician working in this field, but cannot be accepted for the publication in the present form. In particular the description of the mathematical model is totally confused.
The following points must be absolutely reviewed.
0. What is "The 8006 working face". A manuscript should be comprehensible for any reader of the journal.
1. The statement: "In summary, there has been relatively little research into the flow of slurry for grout injection into bed separation. The understanding of slurry flow laws and slurry transport channels is not yet clear. " is unacceptable and has to be changed properly. A lot of work exists in literature on the flow of complex liquids, as e.g. slurry. The authors have to look e.g. for the papers of Prof. Armanini (Trento University) and his research group, as weel as the recent works of Prof. La Rocca et al. (University Roma TRE) on granular flows.
2. The illustration of the mathematical model is very confused. Equations (1), (2) are mass conservation equations of the liquid and the slurry phase respectively. They are not dynamic equations as they do not involve forces. Moreover, why the greek letter tau has been used for the time? Is it a different time from that used in equations (7), (8)?
3. Equations (7), (8) are simply equations (1), (2), written in a different form. They do not add any new information.
4. The flow rate is calculated in terms of the pressure gradient. Is the latter imposed?
5. Equations (10), (11), 12) suddenly appear without no link with the previous ones. Moreover why equation (10), the Navier-Stokes equation, is introduced so that "Slurry transport in large bed separation is simulated using laminar flow"? In order to simulate laminar flows, one should use the Stokes equation. This is a serious theoretical shortcoming.
In conclusion: I suggest that the manuscript is rejected. The authors can work on it and submit it again as a new submission, after that the mathematical model has been totally reviewed and rewritten.